evolution, ecology, genomics

social insect, subsocial, cockroach, major transition, contraction, expansion

**Authors for correspondence:**
Shulin He
e-mail: shulinhe@hotmail.com
Dino P. McMahon
e-mail: dino-peter.mcmahon@bam.de

†These authors contributed equally to this study.

# Evidence for reduced immune gene diversity and activity during the evolution of termites

Shulin He[1,2,3], Thorben Sieksmeyer[1,2], Yanli Che[4],
M. Alejandra Esparza Mora[1,2], Petr Stiblik[3], Ronald Banasiak[2],
Mark C. Harrison[5], Jan Šobotník[6], Zongqing Wang[4], Paul R. Johnston[1,7,8,†]
and Dino P. McMahon[1,2,†]

[1]Institute of Biology, Freie Universität Berlin, Schwendenerstr. 1, 14195 Berlin, Germany
[2]Department for Materials and Environment, BAM Federal Institute for Materials Research and Testing, Unter den Eichen 87, 12205 Berlin, Germany
[3]Faculty of Forestry and Wood Science, Czech University of Life Science Prague, Kamýcká 129, 16500 Prague, Czech Republic
[4]College of Plant Protection, Southwest University, Tiansheng 2, 400715 Chongqing, People's Republic of China
[5]Institute for Evolution and Biodiversity, University of Münster, Münster, Germany
[6]Faculty of Tropical AgriSciences, Czech University of Life Science Prague, Kamýcká 129, 16500 Prague, Czech Republic
[7]Leibniz-Institute of Freshwater Ecology and Inland Fisheries, Müggelseedamm 310, 12587 Berlin, Germany
[8]Berlin Center for Genomics in Biodiversity Research, Königin-Luise-Str. 6–8, 14195 Berlin, Germany

SH, 0000-0003-4817-5646; PS, 0000-0001-6141-5603; PRJ, 0000-0002-8651-4488;
DPM, 0000-0003-1119-5299

The evolution of biological complexity is associated with the emergence of bespoke immune systems that maintain and protect organism integrity. Unlike the well-studied immune systems of cells and individuals, little is known about the origins of immunity during the transition to eusociality, a major evolutionary transition comparable to the evolution of multicellular organisms from single-celled ancestors. We aimed to tackle this by characterizing the immune gene repertoire of 18 cockroach and termite species, spanning the spectrum of solitary, subsocial and eusocial lifestyles. We find that key transitions in termite sociality are correlated with immune gene family contractions. In cross-species comparisons of immune gene expression, we find evidence for a caste-specific social defence system in termites, which appears to operate at the expense of individual immune protection. Our study indicates that a major transition in organismal complexity may have entailed a fundamental reshaping of the immune system optimized for group over individual defence.

## 1. Introduction

Immunity is closely tied with evolutionary transitions, such as the evolution of multicellular organisms from single-celled ancestors and the evolution of eusocial animals from solitary ancestors. The immune system defines the boundaries and threats of biological individuality, and is therefore essential for regulating organism integrity [1]. The evolution of immunity has been well studied at the cell and individual level and efforts to widen understanding to social organisms have been made, such as in bees, thrips and wasps [2–6].

Insect immunity has been studied at multiple levels in a small but growing number of insect models. The insect individual immune system comprises three principle immune pathways: immune deficiency (IMD), Toll and Janus kinase-signal transducer, and activator of transcription [7]. These pathways are typified by pattern recognizing proteins, signalling molecules and effectors,

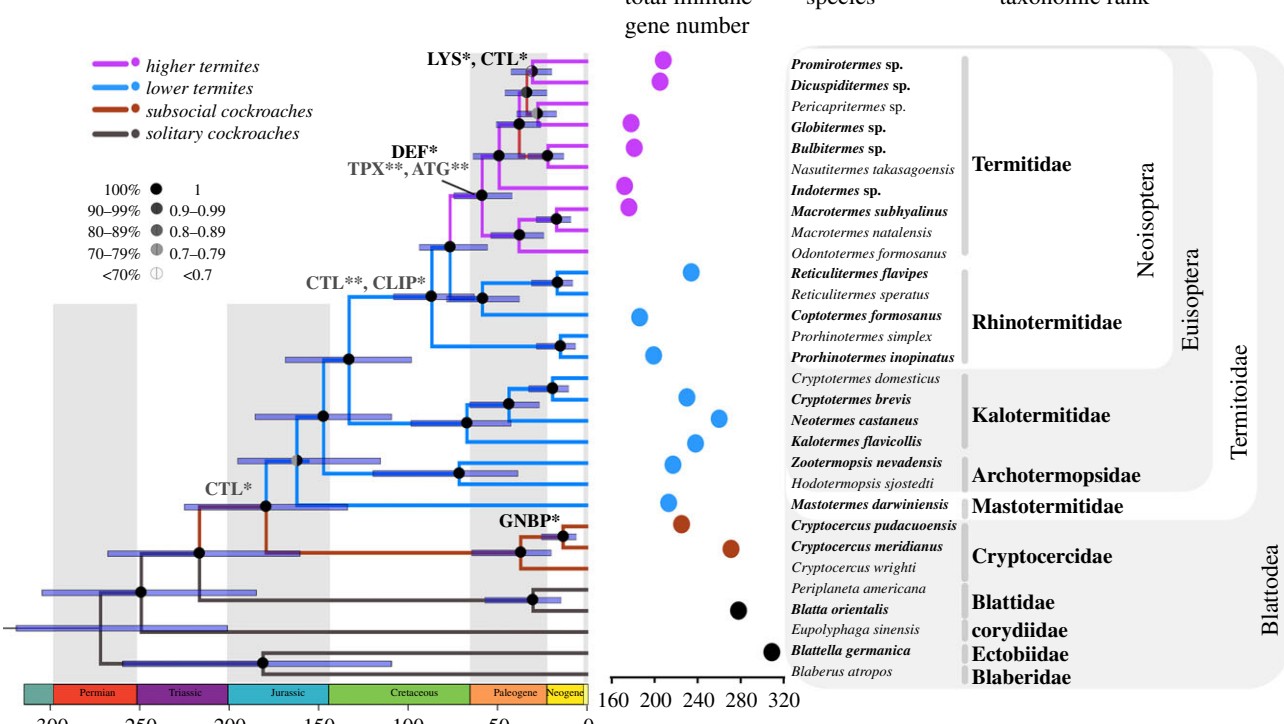

**Figure 1.** Phylogeny of termites and cockroaches alongside total numbers of identified immune genes. Gene family names in grey and black on the phylogeny indicate significant contractions and expansions of individual gene families, respectively. The gene family evolution analysis was conducted in CAFE. ATG, autophagy-related genes; CLIP, serine protease; CTL, C-type lectin; DEF, defensin; GNBP, gram-negative binding protein; LYS, lysozomes; TPX, thiroredoxin peroxidase. Significance levels of 0.05 (*) and 0.01 (**) are shown.

which are responsive to and active against pathogens. In addition to providing protection at the individual level, social insects have developed group-level social immune traits to protect colonies against infection [8,9]. Although social immunity has been the focus of much research in a range of social systems, the evolutionary origins of collective immune defence have received comparatively little attention. As with the evolution of the metazoan immune system, which is thought to have emerged via the co-option of pre-existing molecular modules and functions into novel defensive pathways, it has been hypothesized that social immune systems originated via similar processes [10], with a potentially crucial role for behavioural [11] as well as immune gene adaptations [12,13]. In line with this view, many genes, including immune-related genes, have been shown to display caste-specific expression patterns [14–18]. In addition, enhanced antimicrobial defences have been recorded in some social insects compared with their solitary relatives [2–4,19].

Next to the intensively studied bee, wasp and ant societies, termites represent an essential comparative group due to their ancient and evolutionarily distinct origin of sociality, as well as the presence of extant representatives of the full spectrum of social complexity, including some of the most advanced and ecologically successful societies found on Eearth [20]. Termites possess a rich array of adaptive social immune traits [21], which serve to prevent the spread of infectious diseases within colonies [22]. Genome annotations of termites indicate that they possess a full complement of canonical insect immune gene pathways [23], but a comprehensive analysis of total immune gene family evolution across the full spectrum of termite sociality has hitherto been lacking.

We exploited a transcriptomic approach to compare the immune gene repertoire of 18 cockroach and termite species,

spanning the full spectrum of solitary and social lifestyles (figure 1), including two solitary cockroach species, two species of subsocial Cryptocercus wood-feeding cockroaches and 14 termite species. The two solitary cockroaches, Blattella germanica and Blatta orientalis, represent major subdivisions of the cockroach phylogeny: Blaberoidea and Solumblattodea, respectively [24]. The Solumblattodea is a large monophyletic grouping containing B. orientalis, Cryptocercus and termites. Cryptocercus is a small clade that is robustly supported as the sister lineage to all extant termites. Cryptocercus represents a key lineage in any comparative analysis of termite evolution because it possesses important transitional traits such as subsociality, a wood diet with associated protist gut symbionts, and some developmental similarities with termites [25–27]. The combination of two divergent solitary cockroach species with two unusual subsocial roach species provides a solid baseline for a comparative analysis of termite molecular evolution. The termite species include eight lower termites representing a range of social modes and ecologies and six higher termites belonging to Termitidae which have undergone further transitions in symbiotic and social evolution [28]. Following an investigation into immune gene evolution across a termite phylogeny, we carried out comparative gene expression analyses on representative species bordering the social transition in order to gain deeper insight into the structure of termite immunity.

## 2. Results

### (a) Contractions and expansions of immune gene families in termites

We analysed immune gene evolution over a well-supported termite phylogeny reconstructed from 152 single-copy

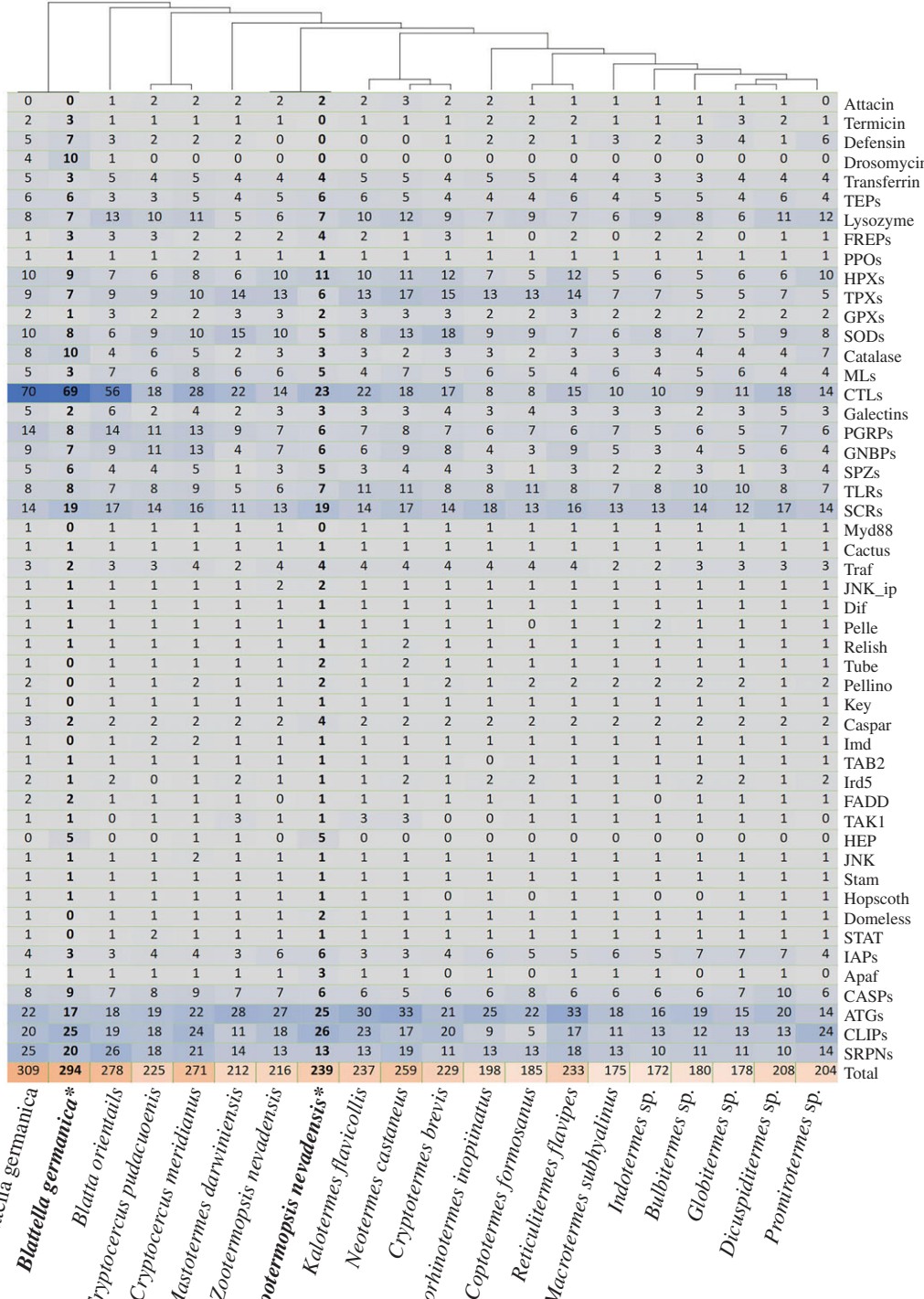

**Figure 2.** Predicted gene numbers in 50 immune gene families. *Gene sets of sequenced genomes were used to verify immune gene predictions from our *de novo* transcriptomic data.

orthologues of 30 cockroach and termite taxa (electronic supplementary material, figure S1). Each gene family was represented in every cockroach and termite species (figure 2), with the noticeable exception of drosomycin, a family of effectors that was lost in termites and wood roaches. An average of 293, 248 and 208 immune-related genes were identified in solitary cockroaches, subsocial wood roaches and social termites, respectively. In a phylogenetic signal analysis, we detected a clear pattern of total immune gene diversity loss during the evolution of termites ($C_{mean} = 0.449$, $p$-value = 0.002; Moran's $I = 0.055$, $p$-value = 0.023; $K = 1.391$, $p$-value = 0.002; $K^* = 0.869$, $p$-value = 0.008; $\lambda = 0.830$, $p$-value = 0.008) with significant positive autocorrelation among species, particularly among immune gene effector and receptor families (figure 2;

electronic supplementary material, figure S2). For example, C-type Lectin (CTL) and peptidoglycan recognition protein (PGRP) genes were notably reduced in number in many termites (figure 2). In a CAFE analysis, two $\lambda$ rates based on clades with a solitary and sub social- or social system were used. The global evolutionary rate of immune gene families in solitary cockroaches (birth/death rate [$\lambda$] = 0.0037) were higher than that of subsocial cockroaches and termites ($\lambda$ = 0.0016). Among effector genes, we find evidence of thioredoxin peroxidases undergoing a contraction in the Termitidae crown group (family wide $p$-value: 0.024, node $p$-value: 0.0013), while defensins underwent an expansion in the same group (family wide $p$-value: 0.011, node $p$-value: 0.0257) (figure 1; electronic supplementary material, figure S3). Of the receptors, CTL

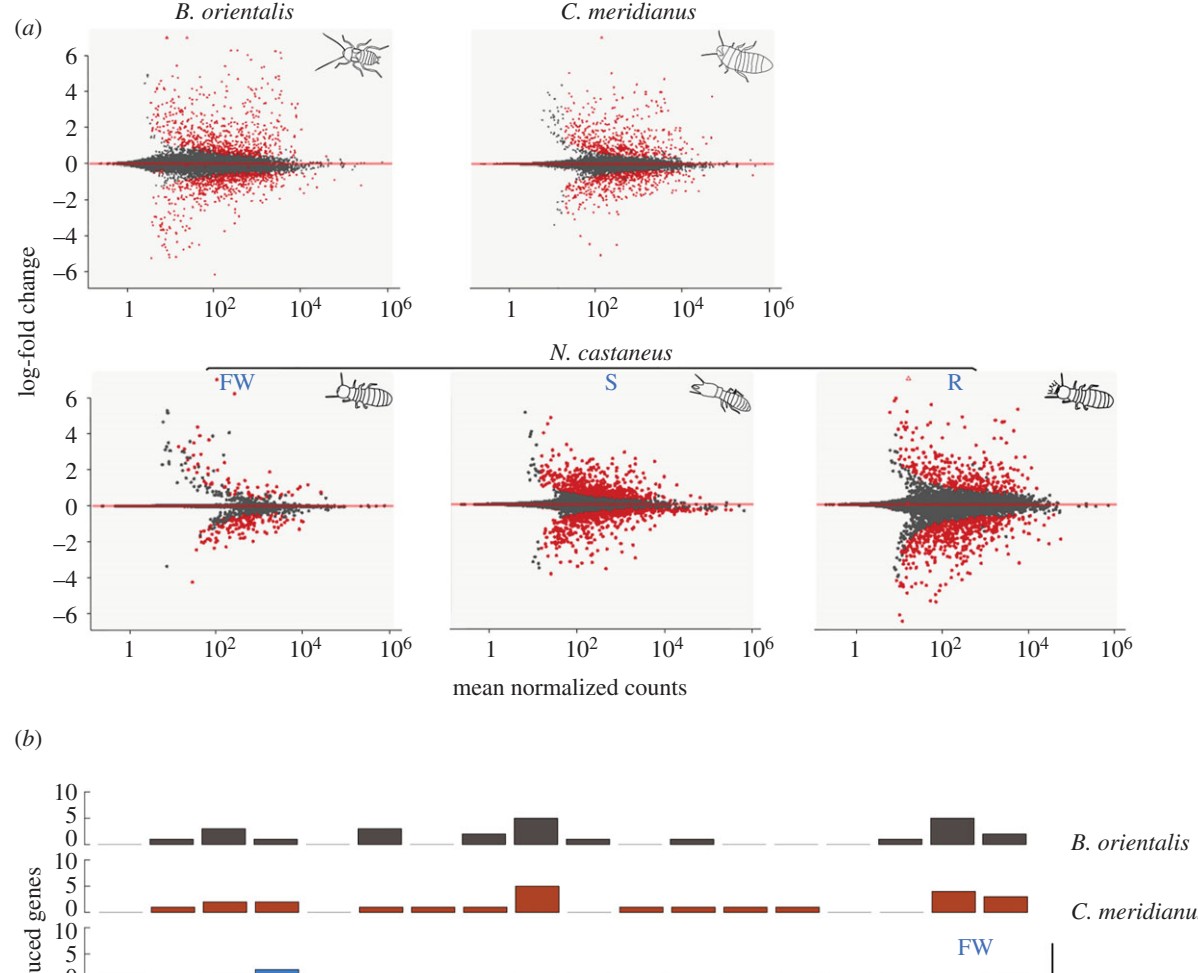

**Figure 3.** Individual immune response following injection with a cocktail of heat-killed microorganisms versus an equivalent Ringer's solution. (*a*) MA plots of gene expression in *B. orientalis* and *C. meridianus* (upper panel) or each caste of *N. castaneus*: FW, false workers; S, soldiers; R, reproductives. Red dots in graphs represent differentially expressed genes. (*b*) Cross-species comparison of total number of significantly induced immune genes following experimental injection. Bars in grey, red and blue represent the solitary cockroach, *B. orientalis*, the subsocial cockroach, *C. meridianus*, and the social termite, *N. castaneus*, respectively. Gene families are categorized from left to right into immune effectors, receptors and signalling genes, respectively.

genes underwent two contraction events during the evolution of termite sociality (family wide *p*-value: 0.011), once in the most recent common ancestor (MRCA) of subsocial wood roaches + social termites (node *p*-value: 0.0173), and once in the MRCA of Rhinotermitidae + Termitidae (node *p*-value: 0.0021). Interestingly, CTL may have also undergone a re-expansion in higher termites (node *p*-value: 0.0278), coinciding with the expansion of lysozymes (family wide *p*-value: 0.02, node *p*-value: 0.0090). We also found evidence of GNBP undergoing an expansion in the common ancestor of subsocial cockroaches (family wide *p*-value: 0.014, node *p*-value: 0.0486) and contractions of CLIP (serine protease) in the MRCA of Rhinotermitidae and Termitidae (family wide *p*-value: 0, node *p*-value: 0.0403) and autophagy-related genes in the MRCA of Termitidae (family wide *p*-value: 0, node *p*-value: 0.0012). Further gene family changes detected at the

tips of the termite phylogeny are listed in electronic supplementary material, figure S3.

## (b) Weak individual immune response in a termite compared with cockroaches

To further investigate the evolution of termite immunity, we compared individual immune responses in a solitary cockroach, *B. orientalis*, a subsocial wood-feeding roach, *Cryptocercus meridianus*, and each caste of the one-piece nesting termite, *Neotermes castaneus*, following direct injection with heat-killed microbes. In the solitary cockroach *B. orientalis*, we found 165 and 263 significantly down- and upregulated genes in immune-challenged individuals, respectively (figure 3*a*). Significantly enriched gene ontology (GO) terms of upregulated genes in *B. orientalis* included Toll and PGRP signalling and

immune/defence processes (electronic supplementary material, table S1). Among total differentially expressed genes, 25 and 10 represented up- and downregulated immune-related genes, respectively (electronic supplementary material, figure S4). In the subsocial cockroach *C. meridianus*, we detected a similar pattern to *B. orientalis*, with 248 and 382 genes being significantly downregulated and upregulated, respectively (figure 3a). Among the total differentially expressed genes, 24 and 19 represented up- and downregulated immune-related genes, respectively (electronic supplementary material, figure S4). Overall, solitary and subsocial roaches are characterized by a significant upregulation of immune genes following immune challenge, including members of several gene families ranging from receptor, effector and signalling molecules in both Toll and IMD immune pathways (figure 3b; electronic supplementary material, figure S4). As in the solitary cockroach, PGRP signalling as well as several immune and defence response categories were significantly enriched in upregulated genes in *C. meridianus* (electronic supplementary material, table S2).

In contrast, a muted response was found in the termites at the caste level, with a reduced number of differentially regulated immune genes as well as non-immune genes across all castes, particularly in false workers, which upregulated only 30 genes in total in response to treatment (figure 3a), which were not significantly enriched for any GO terms (electronic supplementary material, table S3). Upregulated genes in soldiers ($n = 161$) were significantly enriched in immune-related and transport as well as metabolic process GO terms (figure 3a; electronic supplementary material, table S4). Upregulated genes in reproductives ($n = 220$) were significantly enriched in positive regulation of antifungal peptide production and phenol-containing compound biosynthetic processes (figure 3a; electronic supplementary material, table S5). Although total upregulated genes in response to immune challenge were higher in soldiers and reproductives compared to false workers, the number of upregulated immune genes was minimal across all castes, with only 9, 11 and 5 immune genes being significantly upregulated in response to immune challenge in reproductives, soldiers and false workers, respectively. One immune gene, *HPX*, was upregulated across all castes, but most upregulated immune genes were caste-specific and functionally non-overlapping, with reproductives and false workers favouring the upregulation of signalling genes and effector molecules (including an Attacin, a Lysozyme and two *HPX* genes), respectively (figure 3b; electronic supplementary material, figure S5). When summed across castes, the number of significantly upregulated unique immune genes in the termite was similar to the number found in solitary and subsocial roach species (electronic supplementary material, figures S4 and S5). This pattern was also observed for non-immune genes (electronic supplementary material, figure S6). Of the non-immune genes, only two significantly upregulated genes were found to be shared across all three castes. These were a jerky protein homologue-like, and an uncharacterized gene. One gene (poly [ADP-ribose] polymerase 12-like) was significantly upregulated in both false workers and reproductives, while 7 genes were upregulated in both false workers and soldiers and 84 upregulated genes were upregulated in both soldiers and reproductives.

## (c) Caste-specific immunity in the termite *N. castaneus*
We next compared total gene expression differences between castes in the absence of direct immune challenge to understand how caste identity itself shapes constitutive immunity at the individual level. We found that reproductives displayed significantly higher levels of constitutive immune gene expression, followed by false workers, and then soldiers (electronic supplementary material, figure S7 and text). Expression of immune-related genes could be effectively categorized by caste in a principle component analysis (figure 4c). Significantly highly expressed immune genes in reproductives included signalling genes such as Spaetzle, as well as effector molecules Termicin and two Lysozyme genes, while expression of a third Lysozyme, an MD2-like receptor and oxidases were significantly enhanced in false workers. One PGRP gene was significantly highly expressed in soldiers (figure 4d). With respect to differentially expressed genes in general, significantly enriched GO terms of highly expressed genes in the reproductive caste included several reproductive and developmental processes as well as pheromone synthesis (electronic supplementary material, table S6), while carboxylic acid biosynthesis was significantly enriched in highly expressed genes of false workers (electronic supplementary material, table S7). No GO terms were significantly enriched in highly expressed genes of soldiers (electronic supplementary material, table S8).

## (d) Comparison of termite and cockroach gene expression changes in response to a social immune challenge
To explore group responses to a social challenge, we quantified gene expression changes in each caste of *N. castaneus* following colony exposure to immune-challenged nestmates (figure 4a), and compared these with gene expression changes in the solitary cockroach, *B. orientalis*, following group exposure to immune-challenged conspecifics (figure 4b). Individual injection allows us to exclude the pathogen itself as a cue and focus exclusively on the effect of individual health status on social response [29]. In *N. castaneus*, we identified a caste-specific response to social immune challenge, with the following number of differentially regulated genes in each caste (upregulated, downregulated): reproductives (1, 1), soldiers (1, 0), false workers (12, 96) (electronic supplementary material, figure S8). Significantly upregulated genes in false workers were related to metabolic functions and chemoreception, including a fatty acid synthase, a trypsin-like protein, and a gustatory and odorant receptor. Downregulated genes included transport-related, oxidation-related and protease related genes (electronic supplementary material, table S9). In *B. orientalis*, we found a smaller number of genes to be significantly upregulated ($n = 9$) and downregulated ($n = 7$) following exposure to immune-challenged conspecifics. Upregulated genes in conspecifics included 2 serine proteases, a trypsin-4, an ankyrin repeat and fibronectin type-III domain-containing protein 1 as well as 5 other uncharacterized genes. Downregulated genes contained a haemolymph lipopolysaccharide-binding protein (*LPSBP*), a troponin T, a protein obstructor-E and 4 other uncharacterized genes. Upregulated genes were enriched for GO terms linked to serine peptidase and hydrolase activity (electronic supplementary material, figure S8 and electronic supplementary material, table S10), although the role of these genes in cockroach immunity remains unclear.

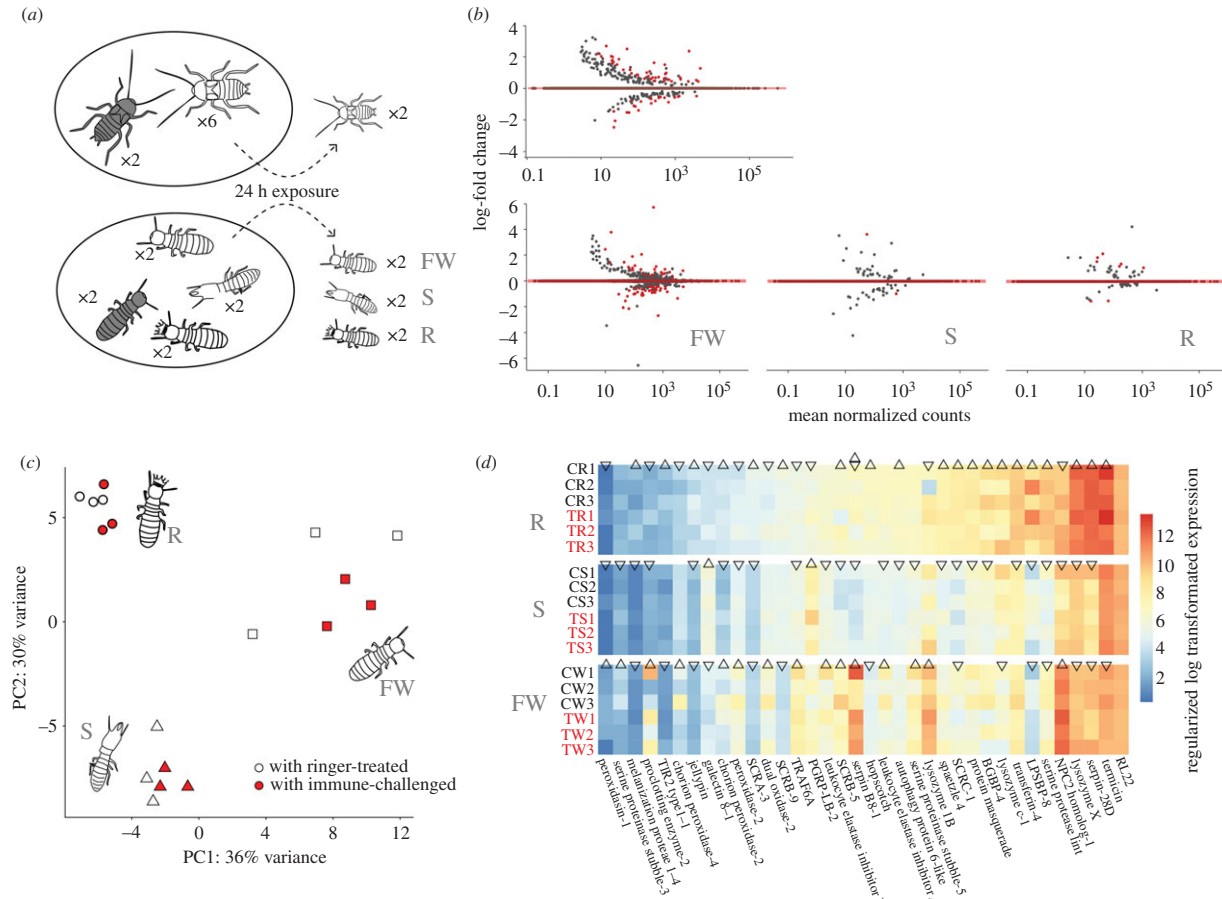

**Figure 4.** (*a*) A diagram of the social group experiment, indicating the design applied to the cockroach *B. orientalis* (upper panel) and the termite *N. castaneus* (lower panel). Individuals marked in grey represent focal individuals. FW, false workers; S, soldiers; R, reproductives. (*b*) MA plots of gene expression in *B. orientalis* conspecifics (upper panel) or each caste of *N. castaneus* nestmates following exposure to treated focal individuals. Red dots in graphs represent the differentially expressed genes. (*c*) Principal component analysis (PCA) of total immune gene expression across all three castes of *N. castaneus* from the social experiment, with points in red indicating social groups exposed to immune-challenged focal individuals. (*d*) A heatmap of differentially expressed immune genes following pairwise comparisons among castes. Expression levels of genes with up-pointing triangles are significantly higher than genes indicated in down-pointing triangles, whereas genes with triangles pointing in the same direction indicate non-significance. Genes marked with both an up- and down-pointing triangle are significantly differentially expressed compared with both other castes, whereas genes lacking a triangle are not significantly differentially expressed compared with both other castes.

## 4. Discussion

The relationship between the evolution of complexity and immunity is attracting attention as researchers increasingly appreciate the interdependency between biological individuality and immunity [1,30]. We explored this relationship in termites by developing a conservative prediction procedure for transcriptomic data to investigate immune gene family evolution during termite social transitions. We detected a full repertoire of immune gene families in all *Cryptocercus* and termite lineages except for the antimicrobial peptide drosomycin. Furthermore, we found evidence for significant contractions of some immune gene families, notably CTL, in early-branching termites followed by potential re-expansions of other immune genes in selected wood roach and termite lineages. For the latter expansions, which depend on subtler changes in gene diversity, high-quality genome sequencing may be required to confirm the observed patterns. We therefore focus the ensuing discussion on the broader patterns that have emerged from our analysis.

It is unclear whether the loss of drosomycin, an antifungal peptide [31], in the ancestor of *Cryptocercus* wood roaches and termites is caused by ecological shifts or the appearance

of social systems, or both. But it is possible that the pleiotropic function of newly evolved fungicidal molecules, like GNBP2 [32], may have led to functional redundancy and subsequent loss of drosomycin.

We found that CTL, comprising a high number of haemolymph LPSBP, underwent two significant contractions in the MRCA of *Cryptocercus* and termites as well as in the MRCA of Rhinotermitidae and Termitidae. CTL is a large group of extracellular carbohydrate binding proteins with various functions [33]. In insects, CTLs contribute to innate immunity and impact infection outcome as well as play a role in host microbiota regulation [34,35]. Evidence exists for the involvement of CTLs in haemocyte nodule formation, encapsulation, melanization and phagocytosis, with mechanisms that are mediated by recognition of pathogen surface molecules, such as lipopolysaccharide, mannose and lipid A, via CTL carbohydrate recognition domains [35,36]. A number of CTLs from cockroaches play a role in immunity, including facilitation of phagocytosis and activation of phenoloxidase activity [37–40]. LPSBPs are significantly expanded in cockroaches [41] and are thought to function as opsonins, by binding specific types of oligosaccharides in the lipopolysaccharide core of invading microorganisms [38,42,43]. Furthermore,

LPSBPs may play a possible function in trapping *Blattabacterium* endosymbionts that have leaked from the fat body into the haemolymph, in addition to functioning in the normal cockroach defence mechanism against foreign microbes [42]. The reduction of CTL genes in *Cryptocercus* and termites suggests a possible reduction in non-self recognition and phenoloxidase activity. The loss of *Blattabacterium* in the ancestor of Euisoptera (all termites excluding Mastotermitidae) may partially explain the pattern of CTL gene depletion, although the significantly reduced diversity of this gene family in both *Cryptocercus* and *M. darwiniensis* indicates that other factors may also be at play.

In bees, immune gene depletion seems to have preceded the evolution of eusociality [5], indicating that immune gene family evolution in Hymenoptera is unrelated to evolutionary transitions in sociality. Although the pattern of immune gene diversity loss in early-branching termites appears to contrast with this finding, the significant expansions of genes, including immune genes, in cockroaches compared to other non-social insects [41] could be interpreted as a relative enhancement of immune gene diversity in the ancestral cockroach clade followed by a return to a more representative level of gene diversity in termites. But given the limited number of solitary cockroach species employed in this study, a full test of this hypothesis awaits the analysis of immune genes from a greater diversity of blattodean lineages. Aside from a general pattern of immune gene diversity loss in termites, our data also suggest a possible re-expansion of immune gene families in some higher termite lineages, potentially resulting from extreme diet diversification and/or shifts in nesting ecology [44]. However, the extent to which these subtler changes in immune family diversity are associated with shifts in termite ecology, microbial symbiosis or sociality requires examination of a wider range of species, as well as using genome sequencing to confirm the *de novo* transcriptome findings. In this light, we urge a general degree of caution when interpreting species-specific gene data until high-quality genomes become available.

The contractions of immune gene families during termite evolution may reflect a general weakening of individual immunity and/or a specialization of immune responses. Similar individual responses to direct immune challenge in the subsocial cockroach *C. meridianus* and the solitary cockroach *B. orientalis* suggest that the initial emergence of subsociality was not associated with significant changes to induced immunity. In contrast, a muted individual immune response across all castes of the lower termite *N. castaneus* indicates that the evolution of termite sociality may be correlated with a reduced ability to mount a robust immune response. A similar phenomenon has been identified in other social insects, including bees and wasps, where eusocial insect groups show weaker melanization responses than their close solitary relatives [45]. This could potentially be the result of trade-offs in selection on individual versus social immunity in more advanced social groups [8].

The social insect colony is a highly organized society with specialized castes. Previous studies in termites have revealed caste-specific expression patterns that reflect functional specialization of castes [14–18]. We found that constitutive immune gene expression is strongly caste specific in *N. castaneus*, reflecting a division of social roles and indicating a significant degree of caste-specific immune defence. For example, constitutive immune gene expression levels were highest overall in reproductives and lowest in soldiers. A similar finding has been reported in comparisons of workers versus reproductives in bees [46] and ants [47] (although see [48]). Due to the limited number of tested termite species in this study, it is difficult to make generalizations about common immune gene expression patterns across all termite clades. Nonetheless, our observations point to a correlation between the evolution of sociality and caste-related immune investment patterns in termites.

Social context plays an important role in coordinating collective behaviour in social insects. It has been demonstrated that caste formation can impact immune gene expression in termites [12,49]. Alongside caste-specific immune gene expression patterns, individuals from different castes may respond to social cues differently, potentially reflecting different levels of investment in individual versus social immunity. Social cues may comprise unique chemical signatures such as cuticle hydrocarbons, which have been shown to be produced by infected worker bees and can evoke an immune response in queens [29]. The termite *N. castaneus* appears able to raise a coordinated caste-specific social immune response, despite this species being a single-piece nesting termite with an intermediate level of social complexity among termites [50]. We recorded a negligible impact of social challenge on soldier and reproductives gene expression indicating that only false workers actively respond to immune-challenged false-worker nestmates, and that they do so by modulating putative sensory and metabolic pathways rather than immune processes. Differentially expressed chemoreception genes in false workers indicate a possible role for chemical communication in coordinating collective social immune responses in *N. castaneus*. However, the importance of behavioural or acoustic cues in termites should be considered as further sources of information in the co-ordination and origins of termite collective defence [51].

We have shown that early-branching termites underwent significant contractions of major immune gene families. Our results reveal a similarity in induced molecular immunity between solitary and subsocial roach species, despite key ecological, developmental and symbiotic traits shared by *Cryptocercus* + Termitoidae [26,27,52]. Termites displayed a comparatively dampened response to direct immune challenge at the caste level. Conversely, termites appear to have evolved caste-specific defences to social as well as individual immune challenge, reflecting a potential change in focus away from individual defence towards group-level protection and fitness.

Our study indicates that the transition to termite eusociality was linked to a significant reconfiguration of termite immune gene diversity and regulation, revealing how a major transition in evolutionary complexity probably entailed fundamental modifications to immune system organization. This study provides a useful comparative foundation for understanding the evolution of termite immunity, which we hope can contribute understanding to the emergence of complexity during this major evolutionary transition in insects.

# 5. Methods

## (a) Insects and microorganisms

Larvae and different castes of nine termite species were extracted from colonies that were kept in the Federal Institute of Materials

Research and Testing (BAM), Berlin, Germany. An additional six species of higher termites were collected from China and Cameroon. Two subsocial wood roaches, *C. meridianus* and *C. pudacuoensis*, were collected from Yunnan, China. The solitary cockroaches, *B. orientalis* and *Blattella germanica*, were kept at 26°C and 75% relative humidity, and were fed with mixed dog food, apples and carrots *ad libitum* until used in experiments. The experimental insects are listed in electronic supplementary material, table S11.

## (b) Sample collection and immune challenge experiments

Freshly cultivated microbes (*Pseudomonas entomophila*, *Bacillus thuringiensis*, *Saccharomyces cerevisiae*) were collected, washed with Ringer's solution, mixed to form a cocktail ($5 \times 10^8$ CFU ml$^{-1}$) and heat-killed at 95°C for 10 min. For the comparative analysis of immune genes, cockroach and termite species were collected by snap-freezing in liquid nitrogen except for the wood roaches which were preserved in RNAlater. Cockroach adults and larvae as well as termites were challenged by injecting with heat-killed microorganisms or piercing with a sterile needle dipped in a heat-killed microbial suspension and collected at 24 h after being challenged. For individual challenge experiments, individuals ($n = 16$ from one cohort of *B. orientalis*, $n = 16$ from 8 colonies of *C. meridianus*, $n = 32$ of each caste from 16 colonies for *N. castaneus*) were weighed and injected with a heat-killed microbial cocktail ($5 \times 10^6$ cells per gram of weight) or Ringer's solution. Following injection, individuals were kept individually with a piece of filter paper. Termites and *B. orientalis* cockroaches were frozen in liquid nitrogen at 24 h following injection, while wood roaches were immersed in RNAlater and stored at −20°C until transportation. For social challenge experiments, *B. orientalis* cockroach groups (containing eight adults from the same ootheca) and *N. castaneus* mini-colonies (containing four false workers, two soldiers and two reproductives from the same colony) were maintained under equivalent conditions. Two false workers from termite mini-colonies and two cockroaches from mini-groups were randomly selected for immune challenge. The focal pairs were injected with heat-killed microbes ($5 \times 10^6$ cells per gram of weight) ($n = 6$) or an equivalent Ringer's solution ($n = 6$). After injection, focal individuals were marked and returned to the group or colony of origin. Every non-marked individual from the termite mini-colonies, or two randomly selected non-marked cockroaches from each group, were frozen in liquid nitrogen at 24 h following the introduction of injected individuals. All samples were preserved at −70°C until RNA extraction. A detailed description of the extraction and library preparation procedures is provided in electronic supplementary material, text.

## (c) Phylogenetic analysis

In addition to sequence data from this study, 10 publicly available transcriptomic datasets were included in phylogenetic inference (electronic supplementary material, table S12). The raw reads were cleaned and filtered before assembled by Trinity (v. 2.5.1) [53]. Raw 454 sequence reads were assembled by using Newbler v. 2.7 (454 Life Sciences/Roche). Subsequently, the assemblies were filtered and translated into proteins by Transdecoder (v. 5.0.1) for ortholog analysis by OrthoFinder (v. 2.0.0) [54]. We selected 152 ortholog groups for constructing the phylogeny using maximum likelihood with RAxML (v. 8.2.12) [55] and Bayesian inference with ExaBayes (v. 1.4.1) [56]. To estimate the divergence times, a molecular clock analysis was performed with PhyloBayes (v. 4.1) [57]. Further details on phylogenetic inference and molecular dating are given in electronic supplementary material, text.

## (d) Immune-related protein identification and evolutionary analysis

The full raw reads from 19 species sequenced for immune gene characterization were assembled with Trinity. Each assembly (except *Pericapritermes* sp., due to low completeness; electronic supplementary material, table S13) was queried against the NCBI nr database by using DIAMOND [58] and annotated by following the guidelines of Trinotate (https://trinotate.github.io/). The proteins of each assembly were predicted by using TransDecoder (v.5.2.0) with a minimum length of 60 amino acids.

We developed a conservative approach to account for potential artefacts associated with the identification of genes from *de novo* transcriptomic data. The strategy exploits cluster information provided by Trinity, which is used to curate the identification of immune genes. The process employs HMMER to identify proteins using a domain-based search strategy. Following filtering, HMMER searches are complemented with trinotate annotations, and assisted by further quality control steps. Briefly, a previously published method [59] was used for the initial quantification of domains containing putative immune functions. Following domain identification, the HMMER output was subjected to a series of stringent filtering steps to exclude misidentified transcripts (electronic supplementary material, text). Following these steps, predicted proteins were queried using blastp against the immune gene family database, and only considered for further analysis when they were assigned to the same immune family as the HMM search. Furthermore, as most genes have multiple immune predicted proteins derived from different isoforms, only one representative isoform was selected. Putative gene targets were further filtered when the output of their predicted proteins from the constructed database did not match the blastp annotation in trinotate. Targets were also removed when their predicted proteins were shorter than 100 amino acids (except antimicrobial peptides). Additional layers of filtering were applied to separate isoforms from paralogues and potential gene fragments based on the headers of trinity assembly output (electronic supplementary material, text). Finally, the expression level of immune genes was examined for all predicted immune genes. Candidate genes with fewer than five counts were manually inspected to verify their identity ($n = 7$). Before using our immune gene predictions in downstream analyses, we tested our method by subjecting our pipeline to the completed genomes of *B. germanica* and *Zootermopsis nevadensis* (aside from the isoform filtering steps), and inspected the level of agreement between our RNAseq data and the equivalent data originating from completed genomes.

The patterns of immune gene evolution over the phylogeny was tested using phylosignal [60]. The expansion and contraction of immune gene families (electronic supplementary material, figure S3) was predicted using CAFE 4.0 (−*p* 0.05) [61]. The details of both analyses are described in electronic supplementary material, text.

## (e) Differential gene expression analysis

The raw datasets for the social and individual immune challenge experiments were assembled together and annotated according to the procedures as applied in the phylogenetic analysis section above. Transcript expression following immune challenge in both experiments was quantified by using salmon [62]. We applied a taxonomy classification with the LCA algorithm in DIAMOND to identify non-target sequences. The transcripts with queried targets from Metazoa were considered as host genes and used for further analysis. Differential gene expression was analysed using the R package DESeq2 [63]. In the comparison of gene expression between termite castes as well as in the individual immune challenge experiments we considered genes to be

significantly differentially expressed when fold changes greater than 4 and adjusted *p*-values < 0.01. Because the responses of nest-mates in the social immune challenge experiment were potentially subtle, we considered genes to be significantly differentially expressed when fold changes greater than 2 and adjusted *p*-values < 0.05. Significantly differentially expressed genes were subject to GO enrichment analysis by using R package goseq with an adjusted *p*-value cut-off of 0.05. The GOs were extracted from the trinotate annotation. After GO enrichment analysis, the redundancy of enriched GOs was reduced by using REVIGO [64].

Data accessibility. All raw data associated with the study are available under BioProject PRJNA635910. The assemblies and quantification data are available from the Dryad Digital Repository: https://doi.org/10.5061/dryad.nzs7h44qp [65]. All codes associated with the study are available at https://github.com/EvoEcoImm/TheEvolutionofTermiteImmunity.

Authors' contributions. D.P.M. and P.R.J. conceived this study. S.H. and T.S. conducted the experiments. S.H., P.R.J., D.P.M. and M.C.H. analysed the data. D.P.M. and S.H. wrote the manuscript. All authors contributed to the writing of the manuscript.

Competing interests. We declare we have no competing interests.

Funding. This study was supported by Freie Universität Internal Research Funding and Devtsche Forschungsgemeinschaft (DFG, grant no. MC 436/5-1) to D.P.M. S.H., P.S. and J.S. are supported by 'EVA4.0' (no. CZ.02.1.01/0.0/0.0/16_019/0000803), and P.S. and J.S. are supported by CIGA no. 20184306. Y.C. and Z.W. are supported by the National Natural Science Foundation of China (grant no. 31672329).

Acknowledgments. We acknowledge the HPC service of the ZEDAT of the Freie Universität Berlin and the Berlin Center for Genomics in Biodiversity Research for sequencing assistance. We thank Jens Rolff, Renate Radek and Sophie Armitage for giving enormous advice and comments. We appreciate Michael T. Monaghan for his advice on phylogenetic analysis.

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
