## [Peer Review File · Proceedings of the Royal Society B: Biological Sciences]

Review History

RSPB-2020-2024.R0 (Original submission)

Review form: Reviewer 1

Recommendation

Accept with minor revision (please list in comments)

Scientific importance: Is the manuscript an original and important contribution to its field?

Good

General interest: Is the paper of sufficient general interest?

Good

Quality of the paper: Is the overall quality of the paper suitable?

Good

Is the length of the paper justified?

Yes

Should the paper be seen by a specialist statistical reviewer?

Yes

Do you have any concerns about statistical analyses in this paper? If so, please specify them explicitly in your report.

No

It is a condition of publication that authors make their supporting data, code and materials available - either as supplementary material or hosted in an external repository. Please rate, if applicable, the supporting data on the following criteria.

Is it accessible?

Yes

Is it clear?

No

Is it adequate?

Yes

Do you have any ethical concerns with this paper?

No

Comments to the Author

Dear authors, you did a great job at demonstrating the evolution of immunity genes in the termite lineage. Your manuscript needs some editing to make it publishable.

Review form: Reviewer 2

Recommendation

Reject - article is scientifically unsound

Scientific importance: Is the manuscript an original and important contribution to its field?

Acceptable

General interest: Is the paper of sufficient general interest?

Acceptable

Quality of the paper: Is the overall quality of the paper suitable?

Marginal

Is the length of the paper justified?

Yes

Should the paper be seen by a specialist statistical reviewer?

No

Do you have any concerns about statistical analyses in this paper? If so, please specify them explicitly in your report.

No

It is a condition of publication that authors make their supporting data, code and materials available - either as supplementary material or hosted in an external repository. Please rate, if applicable, the supporting data on the following criteria.

Is it accessible?

Yes

Is it clear?

Yes

Is it adequate?

Yes

Do you have any ethical concerns with this paper?

No

Comments to the Author

In this paper, the authors investigated evolution of immune gene family diversity in termites by a transcriptomic approach and compared it with those in solitary and subsocial cockroaches. Their main result from the RNA-seq analyses of 14 species of termites, two species of wood-feeding subsocial cockroaches and two species of solitary cockroaches is that immune gene families have experienced significant contractions in the evolutionary process to termites and then a few minor re-expansions in the higher termites.

The story is interesting, but I think that the data they present in this manuscript are weak.

One issue is that they examined only two species for solitary and subsocial cockroaches respectively. I understand it is very difficult to cover all of them, but sample sizes here are so small that it may be difficult to support the broader claims that the authors are making.

Another concern is that they estimated the number of immune gene repertoire by de novo transcriptomic analysis. I understand the authors did hard work, but from my experience, it is very difficult to verify it precisely, especially genes whose expression level is low. The authors insisted on the accuracy of their data by comparing the gene numbers estimated from transcriptome of one cockroach and one termite species with those from their whole genome sequences, but it seems to me that there are considerable differences in number in some gene families.

Again, the bottom line is that the data in this manuscript is not strong enough to support the author's conclusion that the contractions and expansions of immune gene families have actually occurred and played a role in the evolution of termite eusociality.

Meanwhile, I am interested that the gene number of the CTL family has dramatically reduced in termites and subsocial cockroaches. I think that this result is solid, and it would be nice if the authors focus it deeper. There are still so many questions to be answered in the future. What are immunological impacts by the contraction of CTL genes in termites? What is the target of CTLs exactly? Is the phenoloxidase activity actually reduced in termites?

Also, their findings about the caste-specific weak immune responses in termites are interesting. In individual immune challenge experiments, response of immune gene expression in false workers is smaller than those in soldiers and reproductives, whereas in social immune challenge experiments, false workers respond more than soldiers and reproductives. This is a nice contrast, and false workers may have enhanced social immune activity at the expense of individual immune activity to defend their colonies.

Finally, I would like to point out that some figures lack important information probably due to some troubles in the pdf conversion system when the manuscript was submitted.

Here are some specific comments.

Line 95. Is that Fig. S2, but not Fig. S3?

Line 96. I do not understand why the authors included "attacin" genes here. It seems to me that predicted attacin gene number does not change among termites and cockroaches.

Line 165-168. It is not clear why the authors consider that the expression levels of constitutive immune genes are reproductives > false workers > soldiers. They showed a heatmap graph in Fig. S7, but it is not enough to know the order of overall gene expression levels among the castes.

Line 170-174. Gene names are totally missing in Fig 4d.

Line 188. Clarify the figure. Add Fig. S8 after the sentence.

Decision letter (RSPB-2020-2024.R0)

09-Nov-2020

Dear Dr He:

I am writing to inform you that your manuscript RSPB-2020-2024 entitled "Evidence for reduced immune gene diversity and activity during the evolution of termites" has, in its current form, been rejected for publication in Proceedings B.

This action has been taken on the advice of referees, who have recommended that substantial revisions are necessary. With this in mind we would be happy to consider a resubmission, provided the comments of the referees are fully addressed. However please note that this is not a provisional acceptance.

To upload a resubmitted manuscript, log into <http://mc.manuscriptcentral.com/prsb> and enter your Author Centre, where you will find your manuscript title listed under "Manuscripts with

Decisions." Under "Actions," click on "Create a Resubmission." Please be sure to indicate in your cover letter that it is a resubmission, and supply the previous reference number.

Sincerely,
Professor Gary Carvalho
mailto: proceedingsb@royalsociety.org

Associate Editor
Board Member: 1
Comments to Author:

The two main criticisms of the 2nd reviewer need very careful consideration: some justification for why the sample size of cockroaches is sufficient is needed. In an ideal world this is certainly a minimum, but for this kind of study/taxon it is a good starting point; the authors would need to provide further justification and evidence that the small representation of social & nonsocial cockroaches is likely to be a good approximation to those lineages. The second criticism (transcriptome-only) is valid, but such studies can still be very informative for non-model organisms that lack genome sequences. However, the authors need to better acknowledge the short-comings and potential for artefacts, and include some convincing analyses to demonstrate that their conclusions are not compromised by this. Overall, a more measured presentation of the conclusions with tighter interrogation of the data quality is needed.

Reviewer(s)' Comments to Author:

Referee: 1

Comments to the Author(s)

Dear authors, you did a great job at demonstrating the evolution of immunity genes in the termite lineage. Your manuscript needs some editing to make it publishable.

Referee: 2

Comments to the Author(s)

In this paper, the authors investigated evolution of immune gene family diversity in termites by a transcriptomic approach and compared it with those in solitary and subsocial cockroaches. Their main result from the RNA-seq analyses of 14 species of termites, two species of wood-feeding subsocial cockroaches and two species of solitary cockroaches is that immune gene families have experienced significant contractions in the evolutionary process to termites and then a few minor re-expansions in the higher termites.

The story is interesting, but I think that the data they present in this manuscript are weak.

One issue is that they examined only two species for solitary and subsocial cockroaches respectively. I understand it is very difficult to cover all of them, but sample sizes here are so small that it may be difficult to support the broader claims that the authors are making.

Another concern is that they estimated the number of immune gene repertoire by de novo transcriptomic analysis. I understand the authors did hard work, but from my experience, it is very difficult to verify it precisely, especially genes whose expression level is low. The authors insisted on the accuracy of their data by comparing the gene numbers estimated from transcriptome of one cockroach and one termite species with those from their whole genome sequences, but it seems to me that there are considerable differences in number in some gene families.

Again, the bottom line is that the data in this manuscript is not strong enough to support the author's conclusion that the contractions and expansions of immune gene families have actually occurred and played a role in the evolution of termite eusociality.

Meanwhile, I am interested that the gene number of the CTL family has dramatically reduced in termites and subsocial cockroaches. I think that this result is solid, and it would be nice if the authors focus it deeper. There are still so many questions to be answered in the future. What are immunological impacts by the contraction of CTL genes in termites? What is the target of CTLs exactly? Is the phenoloxidase activity actually reduced in termites?

Also, their findings about the caste-specific weak immune responses in termites are interesting. In individual immune challenge experiments, response of immune gene expression in false workers is smaller than those in soldiers and reproductives, whereas in social immune challenge experiments, false workers respond more than soldiers and reproductives. This is a nice contrast, and false workers may have enhanced social immune activity at the expense of individual immune activity to defend their colonies.

Finally, I would like to point out that some figures lack important information probably due to some troubles in the pdf conversion system when the manuscript was submitted.

Here are some specific comments.

Line 95. Is that Fig. S2, but not Fig. S3?

Line 96. I do not understand why the authors included "attacin" genes here. It seems to me that predicted attacin gene number does not change among termites and cockroaches.

Line 165-168. It is not clear why the authors consider that the expression levels of constitutive immune genes are reproductives > false workers > soldiers. They showed a heatmap graph in Fig. S7, but it is not enough to know the order of overall gene expression levels among the castes.

Line 170-174. Gene names are totally missing in Fig 4d.

Line 188. Clarify the figure. Add Fig. S8 after the sentence.

Author's Response to Decision Letter for (RSPB-2020-2024.R0)

See Appendix A.

RSPB-2020-3168.R0

Review form: Reviewer 2

Recommendation

Accept as is

Scientific importance: Is the manuscript an original and important contribution to its field?

Acceptable

General interest: Is the paper of sufficient general interest?

Acceptable

Quality of the paper: Is the overall quality of the paper suitable?

Acceptable

Is the length of the paper justified?

Yes

Should the paper be seen by a specialist statistical reviewer?

No

Do you have any concerns about statistical analyses in this paper? If so, please specify them explicitly in your report.

No

It is a condition of publication that authors make their supporting data, code and materials available - either as supplementary material or hosted in an external repository. Please rate, if applicable, the supporting data on the following criteria.

Is it accessible?

Yes

Is it clear?

Yes

Is it adequate?

Yes

Do you have any ethical concerns with this paper?

No

Comments to the Author

First of all, I agree to the editor's decision and comments to the authors that this study is a first step for understanding evolution of immune gene families in social insects which lack whole genome sequence, and it is also an informative study on such non-model social insects.

I think that the revised manuscript has improved. In the ms, the authors addressed almost all concerns of the reviewer. As for the first concern about the small sample size of solitary and subsocial cockroaches, they provided additional descriptions the reason why the authors chose those species for the analysis and they could be representatives as outgroup species of termites. Although the sample size is a minimum, it may be helpful for readers to address the concern over.

Regarding the second concern about the transcriptome-based analysis, the authors corrected the manuscript in which a several sentences describing controversial results were removed and focused on the convincing results only. The responses are valid, and I am satisfied with it.

I believe that the current form of the manuscript has no significant problems. This study will be very helpful to clarify evolution of immune gene family diversity in termites, especially after future genome sequencing of many termite and cockroach species.

Decision letter (RSPB-2020-3168.R0)

18-Jan-2021

Dear Dr He

I am pleased to inform you that your Review manuscript RSPB-2020-3168 entitled "Evidence for reduced immune gene diversity and activity during the evolution of termites" has been accepted for publication in Proceedings B.

The referee(s) do not recommend any further changes. Therefore, please proof-read your manuscript carefully and upload your final files for publication. Because the schedule for publication is very tight, it is a condition of publication that you submit the revised version of your manuscript within 7 days. If you do not think you will be able to meet this date please let me know immediately.

To upload your manuscript, log into <http://mc.manuscriptcentral.com/prsb> and enter your Author Centre, where you will find your manuscript title listed under "Manuscripts with Decisions." Under "Actions," click on "Create a Revision." Your manuscript number has been appended to denote a revision.

You will be unable to make your revisions on the originally submitted version of the manuscript. Instead, upload a new version through your Author Centre.

- 1) A text file of the manuscript (doc, txt, rtf or tex), including the references, tables (including captions) and figure captions. Please remove any tracked changes from the text before submission. PDF files are not an accepted format for the "Main Document".
- 2) A separate electronic file of each figure (tiff, EPS or print-quality PDF preferred). The format should be produced directly from original creation package, or original software format. Please note that PowerPoint files are not accepted.
- 3) Electronic supplementary material: this should be contained in a separate file from the main text and the file name should contain the author's name and journal name, e.g. `authorname_procb_ESM_figures.pdf`

All supplementary materials accompanying an accepted article will be treated as in their final form. They will be published alongside the paper on the journal website and posted on the online figshare repository. Files on figshare will be made available approximately one week before the accompanying article so that the supplementary material can be attributed a unique DOI. Please see: <https://royalsociety.org/journals/authors/author-guidelines/>

4) Data-Sharing and data citation

It is a condition of publication that data supporting your paper are made available. Data should be made available either in the electronic supplementary material or through an appropriate repository. Details of how to access data should be included in your paper. Please see <https://royalsociety.org/journals/ethics-policies/data-sharing-mining/> for more details.

<http://datadryad.org/submit?journalID=RSPB&manu=RSPB-2020-3168> which will take you to your unique entry in the Dryad repository.

Once again, thank you for submitting your manuscript to Proceedings B and I look forward to receiving your final version. If you have any questions at all, please do not hesitate to get in touch.

Sincerely,
Professor Gary Carvalho
<mailto:proceedingsb@royalsociety.org>

Reviewer(s)' Comments to Author:
Referee: 2

Comments to the Author(s).

First of all, I agree to the editor's decision and comments to the authors that this study is a first step for understanding evolution of immune gene families in social insects which lack whole genome sequence, and it is also an informative study on such non-model social insects.

I think that the revised manuscript has improved. In the ms, the authors addressed almost all concerns of the reviewer. As for the first concern about the small sample size of solitary and subsocial cockroaches, they provided additional descriptions the reason why the authors chose those species for the analysis and they could be representatives as outgroup species of termites. Although the sample size is a minimum, it may be helpful for readers to address the concern over.

Regarding the second concern about the transcriptome-based analysis, the authors corrected the manuscript in which a several sentences describing controversial results were removed and focused on the convincing results only. The responses are valid, and I am satisfied with it.

I believe that the current form of the manuscript has no significant problems. This study will be very helpful to clarify evolution of immune gene family diversity in termites, especially after future genome sequencing of many termite and cockroach species.

Sincerely,
Proceedings B
<mailto:proceedingsb@royalsociety.org>

Decision letter (RSPB-2020-3168.R1)

22-Jan-2021

Dear Dr He

I am pleased to inform you that your manuscript entitled "Evidence for reduced immune gene diversity and activity during the evolution of termites" has been accepted for publication in Proceedings B.

Open Access

Paper charges

Sincerely,

Proceedings B

Appendix A

Associate Editor

Board Member: 1

Comments to Author:

The two main criticisms of the 2nd reviewer need very careful consideration: some justification for why the sample size of cockroaches is sufficient is needed. In an ideal world this is certainly a minimum, but for this kind of study/taxon it is a good starting point; the authors would need to provide further justification and evidence that the small representation of social & nonsocial cockroaches is likely to be a good approximation to those lineages. The second criticism (transcriptome-only) is valid, but such studies can still be very informative for non-model organisms that lack genome sequences. However, the authors need to better acknowledge the short-comings and potential for artefacts, and include some convincing analyses to demonstrate that their conclusions are not compromised by this. Overall, a more measured presentation of the conclusions with tighter interrogation of the data quality is needed.

>>>Response 1: we are grateful to the Editor for the constructive comments. Regarding the first point, we have now provided further description and justification for the cockroach species used, both in the response to reviewer 2 (below) and in the revised manuscript (L74-83). With regard to the Editor's second concern, we are aware of the caveats associated with using transcriptomic data to identify genes. In our revision, we have provided a more explicit description of the comprehensive methodology we have developed to account for the potential artefacts associated with this approach, as well as adding a manual inspection check of lowly represented genes (Main text: L359-377, Supplementary Text: L248-250). Our approach encompasses a range of experimental as well as bioinformatic control measures that are both exhaustive and conservative in nature. We are not aware of further steps that can be implemented to improve upon this so far.

Despite this, we recognize that complete confidence in gene identity for each species may be hard to achieve with transcriptomic data alone. We have therefore re-focused our discussion and conclusions in the revision to highlight the less contentious findings, such as the contractions detected in the CTL gene family (L216-230) and the gene expression section (L269 onwards), while adding greater circumspection to the interpretation of our findings generally in the discussion (L206-208, L240-242, L245-250).

Reviewer(s)' Comments to Author:

Referee: 1

Comments to the Author(s)

Dear authors, you did a great job at demonstrating the evolution of immunity genes in the termite lineage. Your manuscript needs some editing to make it publishable.

>>>Response 2: we thank the reviewer for these comments.

Referee: 2

Comments to the Author(s)

In this paper, the authors investigated evolution of immune gene family diversity in termites by a

transcriptomic approach and compared it with those in solitary and subsocial cockroaches. Their main result from the RNA-seq analyses of 14 species of termites, two species of wood-feeding subsocial cockroaches and two species of solitary cockroaches is that immune gene families have experienced significant contractions in the evolutionary process to termites and then a few minor re-expansions in the higher termites.

The story is interesting, but I think that the data they present in this manuscript are weak.

One issue is that they examined only two species for solitary and subsocial cockroaches respectively. I understand it is very difficult to cover all of them, but sample sizes here are so small that it may be difficult to support the broader claims that the authors are making.

>>>Response 3: We thank the reviewer for their careful reading of the manuscript. Cockroaches and termites are non-model organisms and there are limited genomic and transcriptomic data available for these insects. We regard our study as a first useful foundation for this hitherto understudied but very important group of insects. It provides broad but clearly useful initial insights, as well as helping to formulate data-driven hypotheses that can be further tested in wider genomic studies to come. Crucially, we are confident that the 4 outgroup species we selected are sufficiently diverse to provide a solid baseline for a comparative analysis of termite evolution. Specifically, we selected two representative solitary cockroach species that have been studied in relatively greater detail in other contexts. These species each represent two major clades of the cockroach phylogeny: Blaberoidea (*Blattella germanica*) and Solumblattodea (*Blatta orientalis*), which are distantly related groups within Blattodea (e.g. see Evangelista et al. Proceedings B. 2019). The Solumblattodea is a large monophyletic grouping that contains *B. orientalis*, *Cryptocercus* and termites. This means that *B. orientalis* is much more closely related to *Cryptocercus* and termites than it is to *B. germanica* (Blaberoidea). Similarly, the wood-feeding cockroaches (*Cryptocercus*), which is a small clade of wood roaches sharing transitional traits (e.g social and gut protist) with termites, are robustly supported as the sister lineage to all extant termites. Importantly, *Cryptocercus* is more closely related to termites than to any other cockroach (e.g. see Inward et al. 2007).

Therefore, when we observe shared patterns of gene diversity between these distantly-related, non-monophyletic “cockroach” lineages (i.e. *B. germanica*, *B. orientalis*, and *Cryptocercus*) we can be relatively confident that this is reliable evidence of ancestry, as exemplified by the differences in CTL gene family that we observe between roaches and termites. This is now mentioned in the introduction (L74-83). Furthermore, although the two distant solitary cockroach species display some differences in a minority of immune gene families, on the whole they share similar gene numbers for most families.

Finally, we feel it is essential to highlight the difficulties associated with collecting some cockroach species, particularly the subsocial woodroaches belonging to the genus *Cryptocercus*. These are a rare and very cryptic group of organisms restricted to mountainous areas in eastern Asia and North America. Given their key phylogenetic position as sister group to termites, their inclusion in any analysis of termite evolution is absolutely critical. We invested several months of time trying to collect a sufficient number of individuals to enable their analysis and inclusion in this manuscript. Given this, we believe that our success in gaining 2 *Cryptocercus* transcriptomes (including replicated libraries from a controlled experiment for 1 species) should be viewed as a major achievement in itself. Given

the small size of the genus, this is also pretty good representation of this lineage. On the other hand we do recognize the importance of the general point that the reviewer raises, particularly with respect to the solitary cockroaches, and we have added critical lines to the discussion about the need for greater representation from other cockroach lineages in future studies (L206-208, L240-242, L245-250).

Another concern is that they estimated the number of immune gene repertoire by de novo transcriptomic analysis. I understand the authors did hard work, but from my experience, it is very difficult to verify it precisely, especially genes whose expression level is low. The authors insisted on the accuracy of their data by comparing the gene numbers estimated from transcriptome of one cockroach and one termite species with those from their whole genome sequences, but it seems to me that there are considerable differences in number in some gene families.

Again, the bottom line is that the data in this manuscript is not strong enough to support the author's conclusion that the contractions and expansions of immune gene families have actually occurred and played a role in the evolution of termite eusociality.

>>>Response 4: we appreciate the reviewer's concern and we agree that it can be difficult to predict gene number precisely using de novo transcriptomic data. To account for this, we developed a comprehensive methodology combining a range of experimental as well as analytical control measures. The first control measures involved experimental uniformity, meaning that transcriptomes from all species were prepared in the same way, using equivalent biological material, library preparation and sequencing methods. This is noted in the Supplementary text only (L35-39) due to the strict word limit in the main manuscript. Due to the focus on immune genes, we also prepared transcriptomes from an equivalent pool of immune-challenged and control samples for each species. The aim being to enrich immune genes maximally but uniformly across species. The second control measures involved the development of an exhaustive bioinformatic protocol to attain the most stringent and conservative gene identity pipeline as possible, including a cross-validation step with genomic data from 2 species. During the protocol, we implemented a range of rigorous quality control steps, a detailed explanation of which is given as a summary in the main (Methods: L359-377) and as detailed step-by-step explanation in the Supplementary text. We have also incorporated an additional quality control step of gene expression (L248-250 in the Supplementary Text). We further wish to point to the conservative error model we applied to further mitigate against potential bias in the CAFE analysis (L278-286 in the Supplementary Text). Specifically, we set the error difference to 9, which is a number derived from the comparison of transcriptome versus genome derived gene lists of *Z. nevadensis* and *B. germanica*. It represents the largest difference in gene number that we detected for any single gene family in either species. This conservative estimated error distribution was applied to all species in the dataset. In other words, the maximal divergence observed between transcriptome and genome-derived gene predictions was applied uniformly as an error correction to all species.

>>>Although we have implemented as many mitigation steps as we think are possible, we nonetheless acknowledge the reviewer's general concern that some biases may simply be too hard to identify and correct for. To accommodate this, we have followed the reviewer's suggestion (Response 5 below) and placed much greater emphasis on the CTL result in the revised text (L216-230). At the same time, we are also more circumspect in our interpretation of some patterns we observed in other

gene families (modified abstract, L29-30, deleted text concerning TPX, L214), as well as being careful to explicitly mention caveats that readers should be aware of when interpreting these data (L245-250).

Meanwhile, I am interested that the gene number of the CTL family has dramatically reduced in termites and subsocial cockroaches. I think that this result is solid, and it would be nice if the authors focus it deeper. There are still so many questions to be answered in the future. What are immunological impacts by the contraction of CTL genes in termites? What is the target of CTLs exactly? Is the phenoloxidase activity actually reduced in termites?

>>>Response 5: we agree with the reviewer that the CTL gene family result is indeed fascinating. We have provided deeper information about this family, particularly with regard to the hemolymph lipopolysaccharide-binding proteins (LPSBP), as indicated in lines 216-230. “CTL is a large group of extracellular carbohydrate binding proteins with various functions [35]. In insects, CTLs contribute to innate immunity and impact infection outcome as well as play a role in host microbiota regulation [36, 37]. Evidence exists for the involvement of CTLs in hemocyte nodule formation, encapsulation, melanization, and phagocytosis, with mechanisms which are mediated by recognition of pathogen surface molecules, such as lipopolysaccharide, mannose, and lipid A, via CTL carbohydrate recognition domains [37, 38]. A number of CTLs from cockroaches play a role in immunity, including facilitation of phagocytosis and activation of phenoloxidase activity [39-42]. LPSBPs are significantly expanded in cockroaches [43] and are thought to function as opsonins, by binding specific type of oligosaccharides in the lipopolysaccharide core of invading microorganisms [40, 44, 45]. Furthermore, LPSBPs may play a possible function in trapping *Blattabacterium* endosymbionts that have leaked from the fat body into the hemolymph, in addition to functioning in the normal cockroach defence mechanism against foreign microbes [44]. The reduction of CTL genes in *Cryptocercus* and termites suggests a possible reduction in non-self recognition and phenoloxidase activity. ” We hope that this information, together with the general re-orientation of our manuscript towards this interesting gene family, can help to address the reviewer’s concerns, as well as stimulate further research in this interesting area.

Also, their findings about the caste-specific weak immune responses in termites are interesting. In individual immune challenge experiments, response of immune gene expression in false workers is smaller than those in soldiers and reproductives, whereas in social immune challenge experiments, false workers respond more than soldiers and reproductives. This is a nice contrast, and false workers may have enhanced social immune activity at the expense of individual immune activity to defend their colonies.

>>>Response 6: we thank the reviewer for the comment. The contrasting response of workers under individual and social challenge is indeed intriguing!

Finally, I would like to point out that some figures lack important information probably due to some troubles in the pdf conversion system when the manuscript was submitted.

>>>Response 7: we thank the review for pointing this out and we are sorry for the conversion problem. We have changed the format of figures in the revised manuscript, which we hope help to avoid this problem.

Here are some specific comments.

Line 95. Is that Fig. S2, but not Fig. S3?

>>>Response 8: we thank the review for spotting this error. It has been corrected in the revised manuscript.

Line 96. I do not understand why the authors included “attacin” genes here. It seems to me that predicted attacin gene number does not change among termites and cockroaches.

>>>Response 9: we apologize for the incorrect statement. Compared with cockroaches, attacin genes in termites indeed do not change. However, we find it interesting that the number of attacin genes is consistently around 2-3 in wood feeding cockroaches and most lower termites but only 1 gene in Neoisoptera. This has been corrected in the revised version of the manuscript (L101).

Line 165-168. It is not clear why the authors consider that the expression levels of constitutive immune genes are reproductives > false workers > soldiers. They showed a heatmap graph in Fig. S7, but it is not enough to know the order of overall gene expression levels among the castes.

>>>Response 10: We thank the reviewer for pointing this out. We have run a gene set enrichment analysis to compare the overall expression of immune genes across the 3 castes. We found that immune genes were significantly enriched not only in reproductives compared with false workers and soldiers but also in false workers compared with soldiers. This is now described in the Supplementary text (L378-384) because of the strict word limit in the main text.

Line 170-174. Gene names are totally missing in Fig 4d.

>>>Response 11: we are sorry for the inconvenience because of technical reason during pdf conversion. We have changed the format of figures in the revised manuscript.

Line 188. Clarify the figure. Add Fig. S8 after the sentence.

>>>Response 12: we thank reviewer for spotting the missing clarification. It has been added in the revised manuscript (L185).